# Numerical Simulation of Acoustic Resonance in a Duct Containing a Flat Plate

**Benedict Geihe** 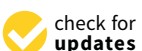**, Jens Wellner \* and Graham Ashcroft**

German Aerospace Center (DLR), Institute of Propulsion Technology, Linder Höhe, 51147 Cologne, Germany; benedict.geihe@dlr.de (B.G.); graham.ashcroft@dlr.de (G.A.)
\* Correspondence: jens.wellner@dlr.de

**Abstract:** This work is about numerical simulations of vortex induced acoustic resonance in a duct, experimentally investigated by Welsh et al. Vortex shedding of low-speed flow over a flat plate excites an acoustic duct mode, leading to a lock-in of shedding and acoustic resonance frequency over a wide range of flow velocities. This study shows that a state-of-the-art compressible Navier–Stokes flow solver is able to capture the lock-in phenomenon. The focus is on the numerical parameters required to precisely recover the experimental results in terms of lock-in range and acoustic pressure levels. Complete and reliable physical data are thereby obtained, which can aid in developing a systematic understanding of the complex flow interactions. Furthermore, hysteresis behavior is discovered and numerically explored.

**Keywords:** CFD; compressible Navier–Stokes; vortex shedding; acoustic resonance; coupling; lock-in

## 1. Introduction

Vortex shedding is an oscillating flow pattern that occurs when fluid overflows a solid bluff body. It depends on flow velocity and the size and shape of the obstacle. Behind the body, vortices detach periodically from either side, forming the well-known von Kármán vortex street. Thereby, alternating low-pressure zones on the downstream side of the structure result in a fluctuating force. Its effect on sound generation has been extensively studied, dating back to the seminal work [1], which introduced the Strouhal number.

When the body is enclosed in a duct, the surrounding air forms a volume that may vibrate by virtue of resonant eigenmodes. Parker experimentally investigated such modes in a hard-walled, low-speed wind tunnel with a cascade of parallel flat plates [2]. The observed pressure fluctuations could be attributed entirely to acoustic effects and their shapes were probed. In [3], they were calculated numerically as solutions of the wave equation under the assumption of a small Mach number. In this work, the two-dimensional $\beta$-mode was defined, a standing acoustic wave with peak pressures at mid-chord position on the duct walls, alternating between above and below the plate.

In the presence of vortex shedding with a frequency close to the eigenfrequency of a duct mode, resonance can occur. In this case, the unsteady flow and the acoustic field mutually interact. Once initiated, the process is self-propelling and ultimately leads to synchronization of both frequencies, known as lock-in. This is accompanied by high sound pressure levels. Cumpsty and Whitehead developed a mathematical model to explain the interaction mechanism [4]. Archibald conducted experiments with acoustic forcing to decouple the fluid dynamic and acoustic parts of the interaction mechanism [5]. Thereby the mathematical model was extended to include a feedback effect. Koch presented a thorough analysis using the analytical Wiener–Hopf solution of an oscillating system [6], covering staggered cascades and subsonic Mach numbers. In [7,8] Welsh et al. investigated the influence of plate geometries on the lock-in range. Furthermore, a potential flow model was derived to predict the actual fluid dynamics leading to resonant conditions.

There are numerous works on the numerical treatment of the described flow phenomenon. Usually incompressible flow and acoustics are treated separately and appropriate modeling has to be applied to include their mutual interaction. For example, Tan et al. used the calculated flow field as an input for the acoustic model [9]. A sinusoidally oscillating velocity perturbation was then applied to mimic the influence of the $\beta$-mode on the flow. The duct mode was obtained from an eigenvalue problem and Howe theory employed to determine if the acoustic resonance could sustain. The resulting scheme proved computationally efficient and allowed for extensive parameter studies.

In spite of the numerous investigations, a unified understanding of the underlying coupling mechanism is not yet available. It is understood that pressure fluctuations, originating from unsteady flow due to the plate, excite and sustain the acoustic resonance. The reverse direction, however, is less clear. The flipping of low and high pressure zones beneath and above the plate is able to trigger separation, both at the leading and the trailing edge. Its effect on sound generation depends on geometric properties of the plate, such as its length and thickness, and the shapes of the leading and trailing edges. To give a few examples, flat plates in open jets were investigated by Parker and Welsh [10]. Among other things, it was found that vortices generated at the leading edge were diffused before reaching the trailing edge when the length to thickness ratio was larger than 25. Hourigan et al. conducted numerical simulations using a spectral-element method [11]. They considered flat plates of varying aspect ratio with square and elliptical leading edges to tell apart the effects of leading and trailing edge vortex shedding and point out their possible interactions. When enclosed in a duct, vortex shedding at the semi-circular trailing edge was identified as the driver of sustained resonance by Welsh et al. in [8]. This is the setup investigated here. For a square leading edge, vortices shed from the leading edge separation bubble were found by Stokes and Welsh in [7]. Experiments with a semi-circular leading edge by Mathias et al. showed a similar, but smaller separation bubble [12]. These works suggest an influence of the acoustic pressure on reattachment. In particular, for low velocity, small vortices in, or fluctuations of the boundary layer would travel along the plate and act as the sound source when passing the trailing edge. Thereby, the number of vortices fitting on the plate gives rise to discrete regimes of excitation. Tan et al. in [9] pointed out the relative phase between the vortices entering the wake and the sound as a crucial element for sustained resonance. Experiments by Katasonov et al. were focused on longer plates [13]. Up to three resonance regimes were found and their dependence on plate length and leading and trailing edge geometry investigated. It was shown that semi-circular edges lead to the strongest acoustic excitation. Furthermore square leading edges were seen to have an influence on wakes, at least with respect to the mean flow state. Finally, analysis of power spectra suggested that the interaction mechanism was truly non-linear.

The motivation for this work is twofold. First it is demonstrated that a state-of-the-art flow solver is able to directly capture the vortex shedding, the acoustic resonance, and their mutual interaction. Second, the obtained results provide highly resolved and accurate temporal and spatial flow field data, which can aid as a first step towards a systematic understanding of the complex flow interactions. Furthermore, a hysteresis behavior is discovered, which has not been reported in detail before.

In contrast to previous numerical studies, this is a high-fidelity approach, which does not make use of any reduced order modeling specific to the test case. Because of the acoustics involved, a compressible flow solver is mandatory. This is challenging because the discrepancy between acoustic and convective velocities results in a badly conditioned numerical problem. Nevertheless, it can be shown that the numerical results agree well with experimental data in terms of predicted shedding frequencies, lock-in range, and sound pressure levels. Chosen numerical parameters are explained and discussed in detail where variations turned out crucial. Previous work on the subject was published by Kociok in [14].

## 2. Methods

All simulations presented in this work have been carried out using the CFD code TRACE. It is a compressible Navier–Stokes flow solver, developed at the Institute of Propulsion Technology of the German Aerospace Center (DLR), with emphasis on turbomachinery flows [15]. A finite-volume discretization of the three dimensional, compressible, unsteady, Favre- and Reynolds-averaged Navier–Stokes equations (URANS) can be solved in steady, unsteady, linearized, and adjoint fashion on structured and unstructured grids. A multi-block approach and modern communication structures allow for highly parallel computations with excellent scaling. Inviscid fluxes are computed using Roe's flux-difference splitting method [16] with Harten's entropy fix [17]. The upwind states are evaluated using the MUSCL family of schemes with several limiters available to smooth the solution in the vicinity of shocks. Viscous terms are discretized using second-order accurate central differences.

The investigated test case was setup to be in close accordance with Welsh et al. [8]. For the computational domain, the working section of the wind tunnel with a length of $L = 2.56$ m and a square cross section of $H = 244$ mm side length was considered. A plate with a thickness of $t = 12.1$ mm and a chord length to thickness ratio of $c/t = 16$ was put in a vertically centered position. Leading and trailing edges were of semi-circular shape with a radius of $r = t/2$. See Figure 1 for a schematic of the setup.

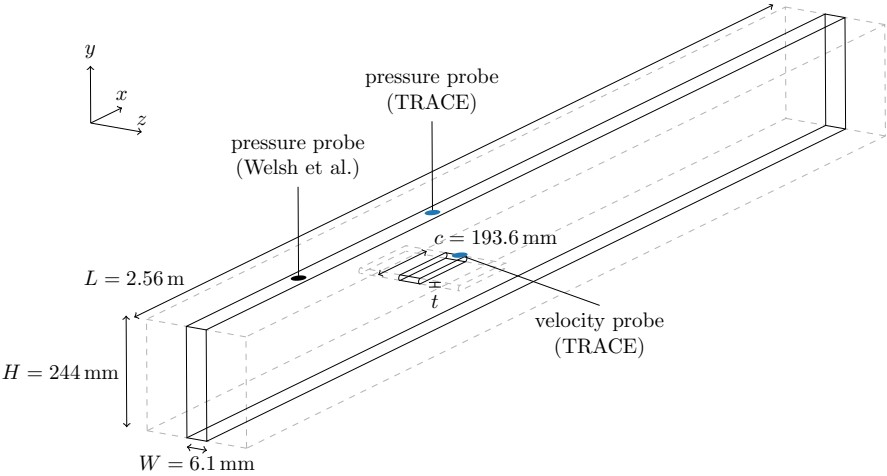

**Figure 1.** Schematic of the experimental and numerical setup.

Of course there are further components of the real experimental apparatus in front and behind, but these should at best not disturb the flow in the working section and are, therefore, ignored in the numerical modeling. For spatial discretization a multi-block structured mesh was generated using the in-house meshing tool PyMesh. Strong mesh refinement was applied towards the plate and the walls to achieve a non-dimensional wall distance $y^+ < 1$. This led to an absolute length of $2 \cdot 10^{-5}$ m for cell edges normal to walls, which also was the global minimum. The resolution in the wake region was chosen rather fine as well. Figure 2 shows an overview and a close-up of the region around the plate.

Owing to the reported two dimensional nature of the flow pattern, the baseline study was confined to a slice of the three-dimensional geometry. In this situation, only one cell was used in spanwise direction, with length $W = 6.1$ mm and inviscid boundary conditions. TRACE recognizes this special configuration and then essentially solves a two-dimensional problem. Altogether, the computational domain of the baseline setup comprised 78,888 cells.

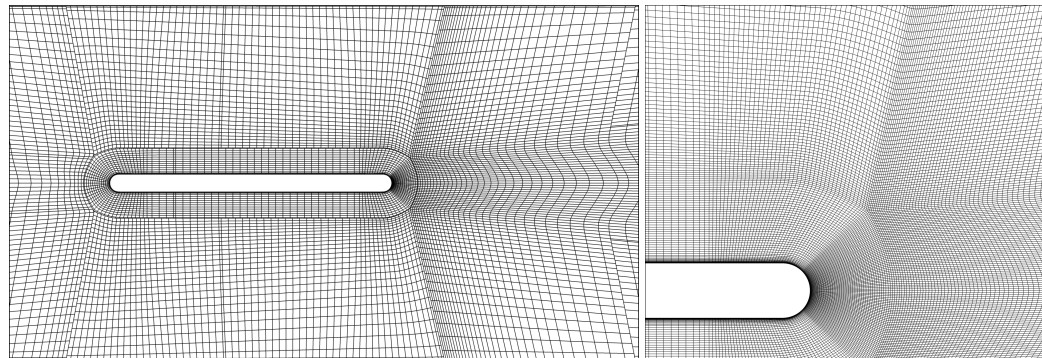

**Figure 2.** Section of the computational domain with every third grid line shown (**left**) and close-up of the region around the trailing edge with every grid line shown (**right**).

The considered flow velocities were in the range of 22 to 37 m/s, corresponding to Reynolds numbers in the range of 17,937 to 30,166 with respect to plate thickness and a viscosity modeled by the Sutherland law. The velocity was set by varying total temperature and pressure at the inflow and static pressure at the outflow. In general, ambient conditions of 293 K and 101,325 Pa were used.

For spatial discretization a third order MUSCL scheme without limiter was used. Temporal integration was done by an implicit Runge–Kutta scheme of third order. The time step size was chosen, such that 128 steps were made during one period of 530 Hz, which was the expected resonance frequency. Consequently, the local CFL number, calculated with respect to the minimal edge length of cells, was about 292 for small cells next to walls when accounting for both convective and acoustic speed. When only the convective speed was considered, the maximal CFL number was about 16 at a distance of several cells diagonally from the leading edge. To solve the system of equations in each step, a dual time stepping, predictor–corrector method was iterated until the residual was smaller than $1 \cdot 10^{-8}$. Simulations were advanced in time until the vortex shedding frequency had settled at a constant value. As acoustic waves were expected to travel along the duct, artificial reflections at the boundaries of the computational domain had to be avoided. To this end, TRACE's non-reflecting boundary conditions [18] were employed at the inflow and outflow panels. For solid walls, non-slip and adiabatic boundary conditions were used. Turbulent effects were modeled by Menter's SST $k$-$\omega$ model. The computer code used double precision floating point numbers.

To obtain the vortex shedding frequency, the vertical velocity component was recorded at a probe located within the wake of the plate, see Figure 1. In a post-processing step, the elapsed times between zero crossings were calculated and averaged. Sound pressure levels were determined from a probe on the upper wall above the plate at mid-chord position. The pressure signal was recorded, centralized, and its root mean square computed. For the sound pressure level, a reference pressure of 20 µPa was used. All of this was performed in an effort to be as close as possible to the experimental setup by Welsh et al. [8]. However, in the experiments, the microphone recording the sound was placed upstream of the plate and the numerical solution of the convected wave equation was used to extrapolate the signal to the described mid-chord wall position.

## 3. Results and Discussion

In this section, first the final numerical outcomes are compared to the experimental results in terms of obtained shedding frequencies and sound pressure levels. After that, the investigation of numerical parameters is summarized, which led to the finally employed setup.

### 3.1. Comparison with Experiment

A series of simulations with different flow velocities ranging from about 22 to 37 m/s were run. The main result is visualized in Figure 3a. Here, the determined shedding frequency $f$ is plotted against the axial inflow velocity $v_x$. As can be seen clearly by the plateau, the lock-in phenomenon is captured by the CFD simulations. At about 23 m/s lock-in occurs, as the frequency suddenly jumps up from the natural shedding frequency to about 520.5 Hz, which is near the expected resonant frequency of about 530 Hz. For increasing velocity, the locked-in frequency rises only slightly until about 33.8 m/s, where it jumps up again and adopts the natural shedding frequency.

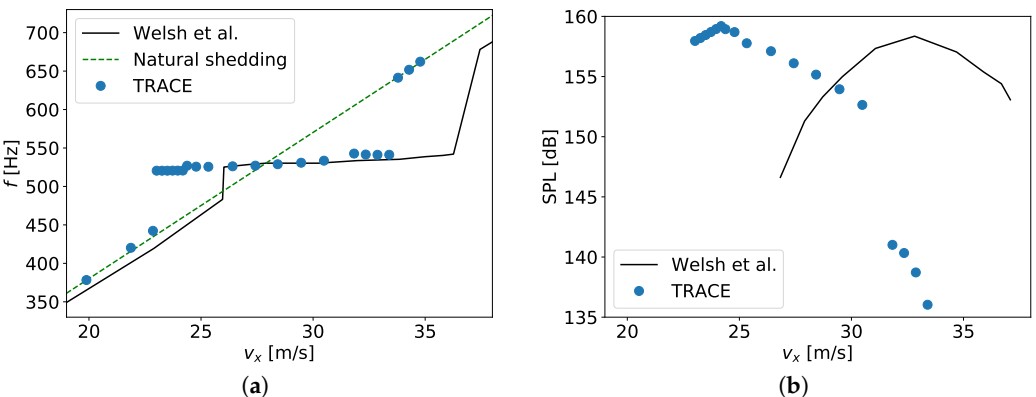

**Figure 3.** Comparison of numerical results to experimental findings by Welsh et al. [8]. (**a**) Vortex shedding frequency; (**b**) Sound pressure level.

When comparing the numerical results to the experimental findings, a horizontal shift of the graphs is apparent. This means that the numerical scheme predicts the onset and cessation of lock-in for lower velocities. We cannot commit ourselves to a definite reason for this deviation. First of all, it could be related to different measuring positions of the inflow velocity. Secondly, the simulated geometry was setup to resemble the experiment, but discrepancies could lead to different behavior in the overall system. An indication is the Strouhal number (Sr). The results correlate best to a natural shedding with Sr = 0.23, as indicated by the dotted fitting line in Figure 3a. On the contrary, the trend in the experimental data indicates a lower value. Thirdly, the aeroacoustics might not be exactly reproduced by the simulations. Thinking of the duct as a resonator, its numerically-found fundamental frequency coincides with the experimental data. However, its response behavior with regard to excitation at lower and higher frequencies differs. Anyway, the lock-in regime spans a range of velocities of about 10.5 m/s and, thereby, almost exactly agrees with the experiment.

The obtained sound pressure levels are shown in Figure 3b. As expected, high values are generally attained where lock-in takes place. The course of values within the range, however, is different. In the numerical results a peak value of about 159.2 dB is found for about 24.2 m/s near the onset of lock-in, whereas in the experiments the peak value of about 158.4 dB is located in the middle of the lock-in region, at about 32.8 m/s. Still the values agree well quantitatively. This is not a matter of course as, often, compressible Navier–Stokes solvers are reported to exhibit erroneous damping in sound wave propagation, at least when low order schemes are used.

The obtained flow field can be seen exemplarily in Figure 4. Four instantaneous snapshots of the fluctuation in the pressure field during one cycle of resonance are shown. In addition to the vortex street, Parker's $\beta$-mode, a standing wave above and below the plate, is clearly visible. Moreover, the radiated sound waves can be seen, which ultimately travel along the duct as evanescent $(1, 0)$-modes. A corresponding video sequence is available in the Supplementary Material.

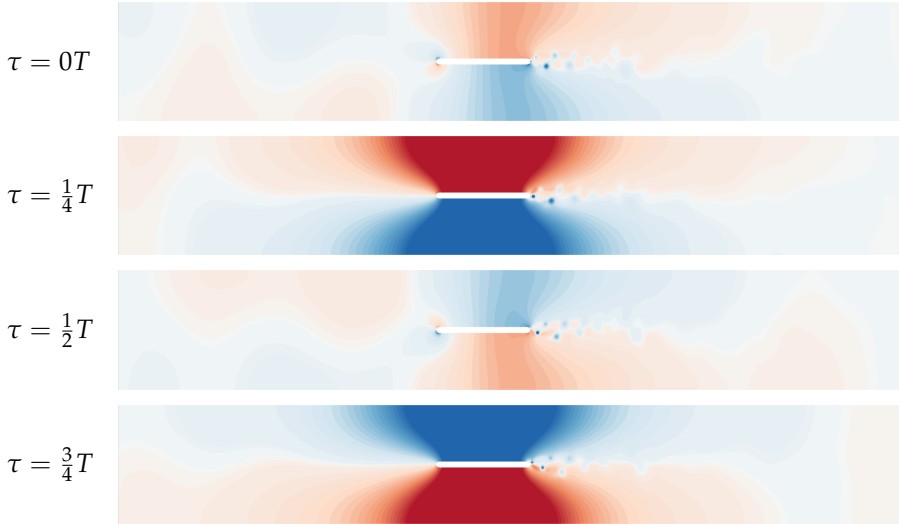

**Figure 4.** Pointwise difference in instantaneous pressure to time averaged solution. Four instantaneous snapshots ($\tau$) during one period $T$ of 530 Hz for a simulation with $v_x \approx 27.4$ m/s are shown. Mapped colors ▬▬ correspond to $[-1000, 1000]$ Pa.

### 3.2. Study of Numerical Parameters

In this section, the investigation of numerical influence factors, which were altered en route to the final result presented above, is summarized. For parameter studies, five simulations with flow velocities of about 23.5, 24.5, 28, 31 and 33 m/s were carried out. These values were carefully chosen to reflect points shortly before and after the beginning, in the middle, and before and after the end of the lock-in region, cf. Figure 3a. The results are assessed based on the obtained axial velocity $v_x$, the vortex shedding frequency $f$ and the sound pressure level (SPL). As a measure of deviation with respect to the baseline setup, the factor $\Delta x = x/x_{\text{baseline}}$ is considered for each quantity.

#### 3.2.1. Initialization and Hysteresis

In the end, the flow solver is comprised of nested iterative schemes, which are not guaranteed to converge to unique solutions. As such, initial values play a crucial role. A natural choice is to start an unsteady simulation from a converged steady state. In Figure 5, an example for a targeted axial flow velocity of 28 m/s is shown.

Figure 5a shows the evolution of the residual during the steady simulation. The unsteady simulation is depicted in Figure 5b–d. To obtain an axial flow velocity of $v_x \approx 28$ m/s, the backpressure was tuned for the steady run. In the course of the unsteady simulation, fluctuations start interacting with the boundary conditions. The axial flow velocity thereby decreases and finally settles in a periodic state, but at a lower, averaged, value. Periodic oscillations can also be seen in the vertical velocity component, probed behind the trailing edge, from which the shedding frequency is determined. The pressure fluctuations at mid-chord position on the wall quickly build up as well and the periodic signal is used to compute the sound pressure level.

When the outlined procedure is followed for all velocities, different results than those presented before in Figure 3a are obtained. Figure 6 shows the high velocity regime, where lock-in is supposed to cease. When the steady state is used as initialization, spurious frequencies appear after lock-in. The measured values actually surpass the natural shedding frequency, which is recovered later on. This seems to be caused by a highly unstable transition regime. As an alternative, the simulations can be started from converged unsteady results for nearby velocities. To this end, two series of computations were conducted, one starting from velocity 30.5 m/s and going up, and one starting from 34.5 m/s and going down. The outcomes are compared in Figure 6.

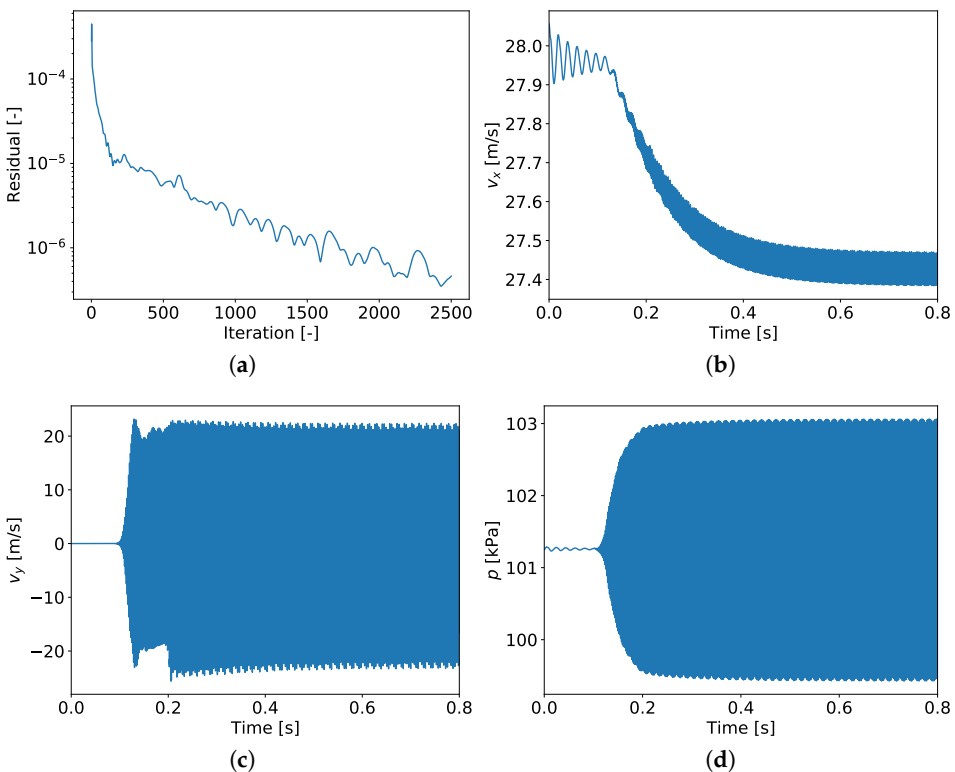

**Figure 5.** Simulation for target velocity $v_x = 28\,\text{m/s}$. (**a**) Residual in steady simulation; (**b**) Unsteady axial flow velocity; (**c**) Unsteady vertical velocity in wake; (**d**) Unsteady mid-chord wall pressure.

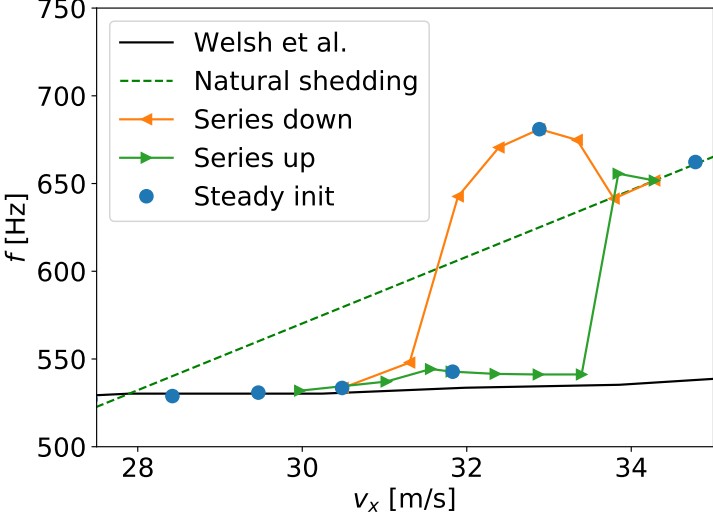

**Figure 6.** Vortex shedding frequency determined from numerical results when initialized by steady states, or by previous results of unsteady simulations for increasing and decreasing velocities comparing with the experimental findings by Welsh et al. [8].

Starting from high velocity, the natural shedding frequency is initially attained. In subsequent simulations, it deviates to higher values, as seen before with the steady initialization. Ultimately, lock-in is reached at about 31.4 m/s. In contrast, simulations started from low velocity continuously maintain the locked state, until a jump towards the natural shedding frequency occurs at about 33.8 m/s. From the investigations it can be concluded that the lock-in phenomenon comes along with a stable state of unsteady flow and an associated hysteresis behavior. Once attained, the locked state is sustained, even if boundary conditions change slightly. A similar phenomenon could be seen by running two series of

computations for the low velocity regime, where lock-in begins. However, differences could only be observed within a far smaller range of velocities. Furthermore, only natural and locked shedding frequencies were found, and no spurious intermediate regime appeared.

Away from the investigated transition regimes, simulations did not show sensitivity with respect to initialization. In view of the experiments by Welsh et al. [8], it is reported that the flow velocity was gradually increased. Thus, the results of the series with increasing velocity were chosen for the final presentation in Figure 3a above.

### 3.2.2. Spatial Resolution

The spatial resolution naturally has a fundamental influence on a numerical simulation. A parameter in the meshing setup was used, which allowed to control the overall cell count, while keeping the geometry and constraints, such as the first wall distance, fixed. Table 1 shows the results for meshing factors 0.5, 0.8, 1, 1.1 and 1.2, where a factor of 1 corresponds to the baseline setup.

As expected, the spatial resolution has significant influence. For a reduction in the overall cell count by 50%, no shedding can be detected anymore, except for the highest velocity considered. Consequently, no sound is predicted either. Using 80% of the baseline cell count is sufficient to obtain good approximations within the lock-in region. However, the onset is missed and the natural frequency still appears for 24.5 m/s. The baseline setup, on the contrary, shows the locked state already at this point. Subsequent mesh refinement does not alter the frequency and sound pressure level significantly. This holds true for the higher velocities as well. Remarkably, mesh refinement also does not change the spurious frequency obtained for 33 m/s. Altogether the findings provide evidence that the chosen spatial resolution is sufficient. In this regard, the numerical modeling is stable, despite the inherently unstable flow regime investigated above.

**Table 1.** Influence of spatial resolution on velocity, shedding frequency, and sound pressure level.

| Meshing Factor | $v_x$ | $\Delta v_x$ | $f$ | $\Delta f$ | SPL | $\Delta$SPL |
|---|---|---|---|---|---|---|
| 0.5 | 23.44 | 1.00 | - | - | - | - |
| 0.8 | 23.36 | 1.00 | 444.7 | 0.98 | 125.4 | 0.98 |
| 1.0 (baseline) | 23.35 | - | 454.1 | - | 127.5 | - |
| 1.1 | 23.33 | 1.00 | 454.2 | 1.00 | 128.1 | 1.01 |
| 0.5 | 24.44 | 1.01 | - | - | - | - |
| 0.8 | 24.34 | 1.01 | 471.8 | 0.91 | 130.2 | 0.82 |
| 1.0 (baseline) | 24.18 | - | 520.7 | - | 159.2 | - |
| 1.1 | 24.17 | 1.00 | 520.7 | 1.00 | 159.4 | 1.00 |
| 0.5 | 27.92 | 1.02 | - | - | - | - |
| 0.8 | 27.42 | 1.00 | 531.9 | 1.01 | 156.2 | 1.00 |
| 1.0 (baseline) | 27.42 | - | 527.1 | - | 156.1 | - |
| 1.1 | 27.42 | 1.00 | 529.0 | 1.00 | 156.4 | 1.00 |
| 0.5 | 30.91 | 1.01 | - | - | - | - |
| 0.8 | 30.48 | 1.00 | 534.2 | 1.00 | 152.4 | 1.00 |
| 1.0 (baseline) | 30.48 | - | 532.9 | - | 152.6 | - |
| 1.1 | 30.49 | 1.00 | 532.9 | 1.00 | 152.6 | 1.00 |
| 0.5 | 32.50 | 0.99 | 537.3 | 0.79 | 151.3 | 1.33 |
| 0.8 | 32.86 | 1.00 | 544.5 | 0.80 | 138.6 | 1.22 |
| 1.0 (baseline) | 32.88 | - | 680.9 | - | 113.7 | - |
| 1.1 | 32.87 | 1.00 | 680.3 | 1.00 | 116.0 | 1.02 |

### 3.2.3. Length of Meshed Duct

The numerical mesh spans the entire duct length of 2.56 m, as described by Welsh et al. [8]. To save computational time, the mesh could be shortened in front and behind the plate. For

a numerical experiment, a reduced length of about 0.92 m was used. The comparison is shown in Table 2.

**Table 2.** Influence of duct length on velocity, shedding frequency, and sound pressure level.

| Duct Length | $v_x$ | $\Delta v_x$ | $f$ | $\Delta f$ | SPL | ΔSPL |
|---|---|---|---|---|---|---|
| baseline | 23.35 | - | 454.1 | - | 127.5 | - |
| short | 23.62 | 1.01 | 461.1 | 1.02 | 128.8 | 1.01 |
| baseline | 24.18 | - | 520.7 | - | 159.2 | - |
| short | 24.37 | 1.01 | 525.0 | 1.01 | 154.9 | 0.97 |
| baseline | 27.42 | - | 527.1 | - | 156.1 | - |
| short | 27.84 | 1.02 | 529.4 | 1.00 | 153.1 | 0.98 |
| baseline | 30.48 | - | 532.9 | - | 152.6 | - |
| short | 31.21 | 1.02 | 545.2 | 1.02 | 139.2 | 0.91 |
| baseline | 32.88 | - | 680.9 | - | 113.7 | - |
| short | 33.29 | 1.01 | 689.4 | 1.01 | 113.6 | 1.00 |

Regarding the regime, in which lock-in occurs, the results agree with the baseline setup. Differences can be seen in the numbers. First of all the settled flow attains different velocities. This is related to the strong pressure fluctuations of the standing wave, which now interfere with the boundary conditions at the closer inlet and outlet panels. A direct comparison of the remaining values is thus less instructive. In any case, there are significant deviations in the sound pressure level. As it is a logarithmic function of the pressure fluctuation, this could be a consequence of the duct being too short to adequately resolve the eigenmode, cf. Figure 4.

### 3.2.4. Full 3D Geometry

A drastic reduction in the cell count in the numerical mesh of the baseline setup was achieved by using only a single cell in the third spatial direction, thereby modeling an effectively two dimensional problem. This is often presumed in the literature, although three dimensional flow was found in the experiments, see (Welsh et al. [8], Figure 5). The meshing setup allowed to discretize the full three-dimensional geometry as well. For this, the end walls were treated in the same way as the walls above and below the plate, with a non-dimensional wall distance of $y^+ < 1$. Overall the cell count thereby rose by a factor of 212. Table 3 shows the comparison to the baseline setup.

**Table 3.** Influence of a fully three-dimensional computational domain on velocity, shedding frequency, and sound pressure level.

| Geometry | $v_x$ | $\Delta v_x$ | $f$ | $\Delta f$ | SPL | ΔSPL |
|---|---|---|---|---|---|---|
| 2D (baseline) | 23.35 | - | 454.1 | - | 127.5 | - |
| 3D | 23.16 | 0.99 | 450.5 | 0.99 | 124.5 | 0.98 |
| 2D (baseline) | 24.18 | - | 520.7 | - | 159.2 | - |
| 3D | 23.91 | 0.99 | 522.0 | 1.00 | 158.7 | 1.00 |
| 2D (baseline) | 27.42 | - | 527.1 | - | 156.1 | - |
| 3D | 27.39 | 1.00 | 525.5 | 1.00 | 157.0 | 1.01 |
| 2D (baseline) | 30.48 | - | 532.9 | - | 152.6 | - |
| 3D | 30.43 | 1.00 | 528.0 | 0.99 | 155.7 | 1.02 |
| 2D (baseline) | 32.88 | - | 680.9 | - | 113.7 | - |
| 3D | 32.71 | 0.99 | 684.6 | 1.01 | 114.0 | 1.00 |

Due to the boundary layers at the end walls, the panel averages at the inflow and outflow boundaries are not expected to exactly match those of the two-dimensional setup. This influences the boundary conditions and leads to differences in the obtained velocities. Nonetheless the overall agreement is very good. In particular, the effective contraction of the channel due to the boundary layers does not influence the lock-in phenomenon significantly. This ex post justifies the reduced two-dimensional setup.

### 3.2.5. Physical Time Integration

To advance the physical time step in the unsteady solver, a third order, implicit Runge–Kutta scheme was used. These schemes are, in general, known to be stable, cf. [19]. The number of time steps used to resolve a period of 530 Hz was 128 in the baseline setup. Simulations with 64 and 256 steps were carried out and the comparison is shown in Table 4.

The number influences the actual solving process. In particular, the employed inner pseudo time stepping scheme requires more iterations to reach a prescribe residual threshold when the physical time step is increased, and vice versa. In addition to this, no significant differences could be observed.

**Table 4.** Influence of physical time step on velocity, shedding frequency, and sound pressure level.

| Time Steps | $v_x$ | $\Delta v_x$ | $f$ | $\Delta f$ | SPL | $\Delta$SPL |
|---|---|---|---|---|---|---|
| 64 | 23.35 | 1.00 | 453.6 | 1.00 | 127.3 | 1.00 |
| 128 (baseline) | 23.35 | - | 454.1 | - | 127.5 | - |
| 256 | 23.34 | 1.00 | 454.6 | 1.00 | 127.6 | 1.00 |
| 64 | 24.18 | 1.00 | 520.5 | 1.00 | 159.2 | 1.00 |
| 128 (baseline) | 24.18 | - | 520.7 | - | 159.2 | - |
| 256 | 24.18 | 1.00 | 520.7 | 1.00 | 159.2 | 1.00 |
| 64 | 27.42 | 1.00 | 528.0 | 1.00 | 156.1 | 1.00 |
| 128 (baseline) | 27.42 | - | 527.1 | - | 156.1 | - |
| 256 | 27.43 | 1.00 | 527.2 | 1.00 | 155.9 | 1.00 |
| 64 | 30.48 | 1.00 | 539.1 | 1.01 | 152.7 | 1.00 |
| 128 (baseline) | 30.48 | - | 532.9 | - | 152.6 | - |
| 256 | 30.49 | 1.00 | 540.7 | 1.01 | 152.4 | 1.00 |
| 64 | 32.89 | 1.00 | 681.7 | 1.00 | 113.0 | 0.99 |
| 128 (baseline) | 32.88 | - | 680.9 | - | 113.7 | - |
| 256 | 32.88 | 1.00 | 680.6 | 1.00 | 113.9 | 1.00 |

### 3.2.6. Pseudo Time Convergence Criterion

The inner iteration of the dual time stepping scheme is controlled by a target residual value for the approximated solution of the system of equations. In previous investigations of vortex shedding from a cylinder in low speed crossflow, this parameter turned out to be crucial, see [20]. As a consequence of the low flow velocity, the problem was ill-conditioned, and the termination threshold had to be set as low as $1 \cdot 10^{-9}$. For the test case investigated here, the velocities were generally higher, and a value of $1 \cdot 10^{-8}$ was used in the baseline setup. Simulations using $1 \cdot 10^{-4}$ and $1 \cdot 10^{-10}$ were also carried out and the results are shown in Table 5.

In fact, making the termination criterion too large cripples the simulations. Vortex shedding does occur, but the frequencies deviate severely. Lock-in is never fully established and sound pressure levels consequently differ as well. A further tightening of the threshold, on the other hand, does not have any influence, besides a much higher required iteration count.

**Table 5.** Influence of termination threshold for the pseudo time solver on velocity, shedding frequency, and sound pressure level.

| Threshold | $v_x$ | $\Delta v_x$ | $f$ | $\Delta f$ | SPL | $\Delta$SPL |
|---|---|---|---|---|---|---|
| $1 \cdot 10^{-4}$ | 23.36 | 1.00 | 373.3 | 0.82 | 125.7 | 0.99 |
| $1 \cdot 10^{-8}$ (baseline) | 23.35 | - | 454.1 | - | 127.5 | - |
| $1 \cdot 10^{-10}$ | 23.34 | 1.00 | 455.2 | 1.00 | 127.7 | 1.00 |
| $1 \cdot 10^{-4}$ | 24.34 | 1.01 | 391.3 | 0.75 | 128.7 | 0.81 |
| $1 \cdot 10^{-8}$ (baseline) | 24.18 | - | 520.7 | - | 159.2 | - |
| $1 \cdot 10^{-10}$ | 24.04 | 0.99 | 520.4 | 1.00 | 159.0 | 1.00 |
| $1 \cdot 10^{-4}$ | 27.70 | 1.01 | 445.2 | 0.84 | 151.5 | 0.97 |
| $1 \cdot 10^{-8}$ (baseline) | 27.42 | - | 527.1 | - | 156.1 | - |
| $1 \cdot 10^{-10}$ | 27.44 | 1.00 | 527.1 | 1.00 | 155.8 | 1.00 |
| $1 \cdot 10^{-4}$ | 30.58 | 1.00 | 448.4 | 0.84 | 158.0 | 1.04 |
| $1 \cdot 10^{-8}$ (baseline) | 30.48 | - | 532.9 | - | 152.6 | - |
| $1 \cdot 10^{-10}$ | 30.51 | 1.00 | 534.2 | 1.00 | 151.9 | 1.00 |
| $1 \cdot 10^{-4}$ | 32.81 | 1.00 | 512.3 | 0.75 | 132.1 | 1.16 |
| $1 \cdot 10^{-8}$ (baseline) | 32.88 | - | 680.9 | - | 113.7 | - |
| $1 \cdot 10^{-10}$ | 32.88 | 1.00 | 680.9 | 1.00 | 113.9 | 1.00 |

### 3.2.7. Pseudo Time Step Size

The pseudo time step size used in the inner iteration is, among other things, controlled by a selectable factor for the local CFL number. Due to the employed implicit time integration scheme, it is generally possible to choose factors much larger than 1. In the baseline setup, 50 was used and additionally 10 and 100 were checked.

The results in Table 6 confirm the central assumption. Neither an increase nor a decrease in the factor, and, thereby, the pseudo time step size, alters the results. Only the number of required inner iterations is affected.

**Table 6.** Influence of the CFL factor on velocity, shedding frequency, and sound pressure level.

| CFL Factor | $v_x$ | $\Delta v_x$ | $f$ | $\Delta f$ | SPL | $\Delta$SPL |
|---|---|---|---|---|---|---|
| 100 | 23.34 | 1.00 | 454.1 | 1.00 | 127.5 | 1.00 |
| 50 (baseline) | 23.35 | - | 454.1 | - | 127.5 | - |
| 10 | 23.35 | 1.00 | 454.5 | 1.00 | 127.3 | 1.00 |
| 100 | 24.18 | 1.00 | 520.7 | 1.00 | 159.2 | 1.00 |
| 50 (baseline) | 24.18 | - | 520.7 | - | 159.2 | - |
| 10 | 24.17 | 1.00 | 520.6 | 1.00 | 159.3 | 1.00 |
| 100 | 27.43 | 1.00 | 527.0 | 1.00 | 156.0 | 1.00 |
| 50 (baseline) | 27.42 | - | 527.1 | - | 156.1 | - |
| 10 | 27.42 | 1.00 | 526.4 | 1.00 | 156.1 | 1.00 |
| 100 | 30.49 | 1.00 | 532.8 | 1.00 | 152.6 | 1.00 |
| 50 (baseline) | 30.48 | - | 532.9 | - | 152.6 | - |
| 10 | 30.49 | 1.00 | 533.2 | 1.00 | 152.6 | 1.00 |
| 100 | 32.88 | 1.00 | 680.9 | 1.00 | 113.6 | 1.00 |
| 50 (baseline) | 32.88 | - | 680.9 | - | 113.7 | - |
| 10 | 32.88 | 1.00 | 679.6 | 1.00 | 114.6 | 1.01 |

### 3.2.8. Turbulence Modeling

The Reynolds numbers in this test case call for an appropriate treatment of fine scale fluctuations, which are not resolved by the computational mesh. Two-equation $k$-$\omega$ approaches were used, namely Menter's SST model [21] in the baseline setup, and the Wilcox model [22] for comparison.

As can be seen in Table 7, both models agree in the low velocity regime, but there is a clear difference for higher velocities. Already at about 30.75 m/s lock-in ceases when using the Wilcox model and, consequently, the sound pressure level is far lower. Interestingly enough, no spurious frequencies, besides the locked and natural shedding frequencies, are found. Instead, natural shedding is immediately recovered.

**Table 7.** Influence of the turbulence model on velocity, shedding frequency, and sound pressure level.

| Model | $v_x$ | $\Delta v_x$ | $f$ | $\Delta f$ | SPL | $\Delta$SPL |
|---|---|---|---|---|---|---|
| Menter (baseline) | 23.35 | - | 454.1 | - | 127.5 | - |
| Wilcox | 23.35 | 1.00 | 454.1 | 1.00 | 127.5 | 1.00 |
| Menter (baseline) | 24.18 | - | 520.7 | - | 159.2 | - |
| Wilcox | 24.18 | 1.00 | 520.7 | 1.00 | 159.2 | 1.00 |
| Menter (baseline) | 27.42 | - | 527.1 | - | 156.1 | - |
| Wilcox | 27.39 | 1.00 | 520.9 | 0.99 | 156.6 | 1.00 |
| Menter (baseline) | 30.48 | - | 532.9 | - | 152.6 | - |
| Wilcox | 30.75 | 1.01 | 578.4 | 1.09 | 132.2 | 0.87 |
| Menter (baseline) | 32.88 | - | 680.9 | - | 113.7 | - |
| Wilcox | 32.73 | 1.00 | 628.8 | 0.92 | 127.9 | 1.13 |

### 3.2.9. Further Parameters

In the course of the investigations, further numerical parameters were considered.

- A wall function approach was used to approximate the boundary layer at the duct wall, instead of using mesh refinement and a non-dimensional wall distance $y^+ < 1$;
- The modeling for compensation of the stagnation point anomaly was deactivated, and also changed to the Schwarz approach, see e.g., [23], instead of the default according to Kato-Launder [24];
- Limiting functionals, in particular Van Albada's, were activated;
- Menter's reattachment modification was activated in the turbulence model;
- A backwards difference scheme of second order (BDF2) was used, instead of the Runge–Kutta scheme;
- Farfield boundary conditions were tested, as used by Kociok in [14];
- Multiple cell layers, instead of just one, were used in spanwise direction for the slice of the three dimensional geometry.

None of those parameters had an impact on the results. Using inviscid boundary conditions for the duct walls changed the flow velocities and thereby the lock-in range.

In conclusion, the spatial resolution, the termination threshold of the inner iterative solver in each physical time step, and the turbulence model were identified as numerical parameters with significant influence. The respective variations in the results provided evidence of a consistent and reliable final numerical setup. Above all, initialization turned out crucial because of the discovered hysteresis.

## 4. Conclusions

In this work, a classical test case was investigated, showing acoustic resonance in a duct caused by vortex shedding. It was demonstrated that a state-of-the-art flow solver is able to capture the complex interplay of aerodynamics and aeroacoustics, and their mutual frequency locking. In comparison to experimental data, the onset and cessation of

lock-in were predicted for lower velocities. The spanned range for lock-in and the resonant frequency however agreed well. Sound pressure levels were predicted in the correct range of values, but showed a different course within lock-in. The deviations indicate that, for reasons unknown so far, the numerical modeling led to a different behavior of the resonating system. Numerical parameter studies proved the validity of the employed CFD setup. Apart from spatial grid resolution, pseudo time convergence criteria, and turbulence modeling, the initialization turned out crucial and revealed hysteresis effects at the borders of the lock-in range not reported in detail before. Altogether, highly resolved and accurate temporal and spatial flow field data were obtained, which can aid in further studies of the mutual interaction phenomenon.

**Supplementary Materials:** The following supporting information can be downloaded at: https://www.mdpi.com/article/10.3390/fluids7080253/s1, Video S1: video sequence corresponding to Figure 4.

**Author Contributions:** Conceptualization, B.G. and G.A.; Data curation, J.W.; Formal analysis, B.G.; Funding acquisition, G.A.; Investigation, B.G.; Methodology, B.G.; Project administration, G.A.; Resources, J.W.; Software, B.G., J.W. and G.A.; Supervision, G.A.; Validation, B.G., J.W. and G.A.; Visualization, B.G.; Writing—original draft, B.G.; Writing—review and editing, B.G., J.W. and G.A. All authors have read and agreed to the published version of the manuscript.

**Funding:** During work on the subject the first author was supported by the German Federal Ministry of Economic Affairs and Energy within the scope of the fifth call of the Federal Aviation Research Programme under grant number 20T1518B.

**Data Availability Statement:** The data presented in this study are available on request from the corresponding author. The data will be made publicly available in an institutional repository in the future.

**Conflicts of Interest:** The authors declare no conflict of interest.

## Abbreviations

The following abbreviations are used in this manuscript:

| | |
|---|---|
| CFD | Computational Fluid Dynamics |
| CFL | Courant–Friedrichs–Lewy |
| DLR | German Aerospace Center |
| MUSCL | Monotonic Upstream-centered Scheme for Conservation Laws |
| SPL | Sound Pressure Level |
| SST | Shear Stress Transport |
| URANS | Unsteady, Favre-, and Reynolds-averaged Navier–Stokes |

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
