# Peer review of "Numerical Simulation of Acoustic Resonance in a Duct Containing a Flat Plate"

_fluids, doi:10.3390/fluids7080253_

Round 1
Reviewer 1 Report
Please refer to the attached review report

Reviewer 2 Report
This is a very good paper.
The reason for the "Average" rating for Significance of Content is because the paper addresses just one specific geometry. But, the authors have done an excellent job in addressing that geometry.
The paper is very well written, easy to follow and clearly explains what the authors did and why they did it. I liked the carefully parameter investigation.
I must admit that I did not fully understand Fig. 3 until I saw the supplementary video. Now it makes excellent sense.
In my opinion, the paper is ready to be published.
Reviewer 3 Report
This paper reports the numerical investigation of oscillating flow around a flat plate in a duct. To simulate the interaction between the flow fluctuation around the plate and the acoustic resonance in the duct, a compressible flow simulation was conducted by utilizing the unsteady RANS technique. The parameters of the numerical method were widely explored to compare with the experiments, and these results are worth reporting as a research paper in the Journal Fluids. However, the reviewer suggests the following things for the authors to reconsider and revise before the publication.
1. In the introduction, the authors explain the previous studies reporting the fluids around the flat plates. However, it is unclear what is still unsolved or unclear and what the authors want to clarify in this paper. The authors declare that the aim of the study is to demonstrate that their flow solver can capture the flow-acoustic interactions. However, there are already lots of solvers that can simulate the flow-acoustic interactions in low Mach number flows, and there is nothing new about that. The authors should clearly state the purpose of this paper.
2. For the section headings, Section 2 should be just “Methods” since there is no description of "materials" in the numerical study. In addition, the authors wrote lots of discussion in the results section (Section 3), and the discussion section (Section 4) is too short. Section 3 should be “Results and discussion” and Section 4 should be “Conclusion”.
3. In the second paragraph of section 2, it would be helpful for the reader if there is a schematic of the duct and flat plate with each dimension. Also, please provide the radius of the edges of the flat plate. Then, indicate the probe location of the pressure for the results of the simulation and experiment in the figure.
4. The authors describe that only one cell was used in the spanwise direction with inviscid boundary conditions. Does this mean the spanwise grid size is 244 mm? What is the inviscid boundary condition? Does it work like a periodic boundary or symmetric boundary condition?
5. Please provide the minimum grid size (Fig. 1) and related CFL number (at least for the baseline) in the method section. And also, if the simulation considers the acoustic interaction, the CFL number should be calculated based on the speed of sound.
6. What is the limiter and Harten entropy fix constant of MUSCL scheme? Adding a reference would be helpful for the readers.
7. The authors describe that they chose the dimensions of duct length 2.56 m with a non-reflecting boundary condition to reproduce the experimental conditions of Ref [1]. However, there should be a settling chamber and a fan upstream of the duct in the experimental setup, and the exact non-reflection boundary cannot be reproduced in the experiment. The authors should carefully describe the difference in the boundary conditions between the simulation and experiment.
8. What kind of boundary condition was used for the walls of the duct and plate? (non-slip and adiabatic?)
9. In the result, the lock-in frequency was observed at around 520 Hz. However, how was this frequency determined? If the resonance frequency is determined from the eigenmodes, there should be a resonance wavelength and it can be predicted by an equation like Parker (1967)1. If the authors could describe this mechanism, the authors might be able to describe the details of the discrepancy in lock-in velocities between the experiment and simulation.
10. Why did the authors sample the pressure from a different place from the experiment? If the previous study used the wave equation to extrapolate the signal at the mid-chord wall position, the authors can do the same thing to compare.
11. At the end of the third paragraph in section 3.1, the authors stated that compressible Navier-Stokes solvers show erroneous damping in the sound wave propagation. However, this damping is probably caused by the relatively low accuracy of the schemes in this study. For the general fluid-acoustic interaction in low Mach number flows, sixth or more higher-order accuracy schemes are often used to solve both acoustic and fluid pressure fluctuations, and there should not be the “erroneous damping” in the appropriate compressible solvers.
12. The symbol ∆ in the tables shows the difference between the baseline values? If so, please indicate that. In addition, it is a bit confusing that the authors compared the baseline values with the baseline and wrote 1.00 in all cells of baseline. Those values should be removed from the tables.
Reference
1 R. Parker, “Resonance effects in wake shedding from parallel plates: Calculation of resonant frequencies,” J. Sound Vib. 5(2), 330 (1967).
Round 2
Reviewer 1 Report
The manuscrip has been improved significantly and all my queries/comments have been addressed properly. I have no further queries and would like to recommend publication of the paper.